# Analysis of Mock Conversations Across Large Language Models

## Abstract

The rapid advancement of large language models (LLMs) has enabled increasingly sophisticated conversational agents, and systematic comparisons of their conversational behaviors are of great importance. In this study, we generated mock conversations between two people using four LLMs—ChatGPT Free version without account, Gemini-2.0-flash, GPT-5 thinking model, and Claude Opus 4.1—prompted to produce 30-turn interactions each. We quantitatively analyzed multiple conversation-level features, including structural metrics (e.g., number of turns, utterance length), lexical and linguistic properties (e.g., type-token ratio, noun/verb ratios, lexical alignment), sentiment and emotion, repetition and novelty, question-response patterns, speaker balance, and linguistic complexity measured via perplexity. Statistical tests (Kruskal-Wallis), feature importance analyses using random forests, and dimensionality reduction (PCA) were employed to identify discriminative features and uncover patterns across models. Results revealed that GPT-5 exhibited high novelty, lexical diversity, and complexity but shorter utterances, whereas ChatGPT Free produced longer, more positive utterances with higher question rates. Claude Opus 4.1 generated the longest conversations with balanced linguistic profiles, and Gemini-2.0-flash was generally intermediate. Our work provides a multi-dimensional understanding of AI conversational behavior within single-agent interactions. This can offer insights into model selection, fine-tuning, and the design of future human-AI dialogue systems.

## 1 Introduction

The rapid evolution of conversational AI agents has significantly impacted various domains. Recent advancements have led to the development of different models, each exhibiting unique conversational characteristics. Understanding these differences is crucial for selecting the appropriate model for specific applications. There are many studies dealing with AI and conversation. For example, Ebubechukwu, et al. (2024) generated artificial conversations using GPT-4o and compared human and GPT-4 evaluations of predefined key performance indicators. Xu, et al. (2024) discussed LLMs in the context of second langage learning and evaluated their dialogues based on response success rate, suggestion success rate, and session success rate. In addition to these, there are a large number of evaluation methods for LLMs in multi-turn conversational settings (Guan, et al. (2025)). Rządeczka, et al. (2025) discussed conversational AI in mental health interventions, especially in the context of cognitive biases. Anderson, et al. (2025) discussed LLM-associated words commonly used by LLMs and pointed out that they might be shifting our language patterns. These studies from multiple perspectives and contexts show a growing interest in AI conversation patterns.

To better understand conversations generated by LLMs, in this study, we systematically evaluate and compare the conversational characteristics of four LLMs; ChatGPT Free version, Gemini-2.0-flash, GPT-5 thinking model, and Claude Opus 4.1. Our analysis encompasses various aspects, including

conversation structure, lexical and linguistic features, sentiment and emotion, repetition and novelty, question and interaction patterns, speaker balance, and complexity. By employing statistical tests, feature importance analysis, and dimensionality reduction techniques, we provide a comprehensive overview of how these models differ in their conversational behaviors. The findings from this study offer valuable insights for developers, researchers, and practitioners seeking to understand the nuanced differences between contemporary conversational AI agents. By highlighting these distinctions, we aim to inform the selection and optimization of AI models for specific tasks, ultimately enhancing user experience and application efficacy.

## 2 Methods

### 2.1 Conversation Generation

We generated a synthetic dataset of multi-turn conversations by prompting four large language models (LLMs): ChatGPT (Free version without account), Gemini-2.0-flash, GPT-5 (thinking model), Claude Opus 4.1

Each model was prompted to produce five independent conversations, with 60 utterances per conversation (30 per speaker). We used the following controlled prompt to standardize conversation length and format:

> Please make a mock conversation between two people A and B. The output format should be as follows: conversation = [ ("A", "XXX"), ("B", "YYY"), ("A", "ZZZ"), ("B", "WWW"), ] Please make A and B speak 30 times each.

### 2.2 Feature Extraction

We computed a rich set of linguistic, semantic, and interactional features for each conversation.

Libraries and models used in this study was as follows:

- spaCy (en_core_web_sm): tokenization, POS tagging, sentence segmentation
- SentenceTransformers (all-MiniLM-L6-v2): utterance embeddings, cosine similarity for topical consistency
- TextBlob: sentiment polarity
- HuggingFace Transformers (GPT-2): token-level perplexity for measuring predictive difficulty

For each conversation, we extracted:

**Basic statistics:** Number of turns (`num_turns`), mean utterance length (`avg_len`), type-token ratio (`ttr`), per-speaker word balance

**Syntactic features:** Question rate (fraction of utterances ending with "?"), noun-to-token ratios, verb-to-token ratios, average sentence length

**Semantic consistency:** Topical consistency (mean cosine similarity between consecutive utterances), embedding variance (dispersion of utterance embeddings), novelty (complement of topical consistency, reflecting informational change)

**Discourse dynamics:** Self-repetition rates, partner word reuse rates, lexical alignment between speakers (shared vocabulary proportion), lexical entropy (Shannon entropy over word distribution), lexical convergence over conversation progression

**Pragmatic and affective features** Mean of sentiment polarity, variance of sentiment polarity, sentiment shift between first and second halves, pattern rate (frequency of common conversational clichés), questionresponse matching rate (proportion of questions followed by an answer), pronoun balance (I vs. you ratio)

**Complexity:** Average utterance perplexity under GPT-2 (an estimate of predictability)

This feature set captures a diverse set of conversation properties, from lexical richness to semantic coherence and pragmatic interaction quality.

## 2.3 Statistical Analysis and Visualization

**Exploratory Visualization:** To compare distributions across agents, we produced box plots of each feature grouped by agent, feature correlation heatmap (Pearson correlations across all features), and Principal Component Analysis (PCA) with two components, to visualize overall separation between agents in a reduced feature space. These visualizations allowed qualitative assessment of whether certain features systematically varied between LLMs.

**Inferential Statistics:** For each feature, we performed a KruskalWallis H-test (a non-parametric test appropriate for small samples) to assess whether feature distributions differed significantly across agents. If a feature contained fewer than two unique values, it was excluded from testing. Test statistics and p-values were recorded and sorted by significance.

**Feature Importance Analysis:** To identify which features most strongly discriminated between agents, we trained a Random Forest classifier (scikit-learn, `random_state=0`) using standardized feature vectors (StandardScaler). Feature importance scores were extracted and visualized as a ranked bar plot.

## 2.4 Computational Resource

All analyses were implemented in Python 3.10. The analysis was conducted on a server equipped with an Intel Xeon Gold 6130 CPU 2.10GHz, three NVIDIA Quadro GV100 GPUs, and 256 GB of RAM. Also, we placed our code on Google Colaboratory (see Reproducibility Statement for the link) for easier reproducibility. We were able to extract features within 5 minutes on Google Colaboratory without GPU. All the other analyses were almost immediate.

# 3 Results

## 3.1 Patterns Observed from Exploratory Visualization

We generated box plots of each feature grouped by agent (Figure 1).

**Conversation Structure:** Analysis of conversation length and utterance properties revealed systematic differences across agents. Claude Opus 4.1 produced the longest conversations in terms of number of turns, while ChatGPT Free exhibited the widest variability. In contrast, average utterance length and average sentence length were highest for ChatGPT Free, whereas GPT-5 thinking model tended to produce shorter utterances and sentences. These findings suggest that ChatGPT Free emphasizes longer, more elaborate responses, while GPT-5 favors conciseness, and Claude maintains extended but balanced interactions.

**Lexical and Linguistic Features:** Lexical diversity and syntactic patterns differed notably between agents. GPT-5 thinking model exhibited the highest typetoken ratio (TTR) and novelty, indicating more varied and creative language, along with a higher noun ratio but lower verb ratio, suggesting noun-heavy phrasing. In contrast, ChatGPT Free and Claude Opus 4.1 displayed more balanced syntactic distributions. Claude Opus 4.1 also showed the highest lexical entropy, reflecting broader word usage, while lexical alignment and convergence were elevated for ChatGPT Free and Claude, indicating stronger mirroring of conversational partners.

**Sentiment and Emotional Patterns:** Sentiment analysis revealed distinct affective profiles. ChatGPT Free and Claude Opus 4.1 tended toward more positive sentiment, whereas GPT-5 thinking model had lower average sentiment and demonstrated more negative sentiment shifts over the conversation. Variation in sentiment was slightly lower for GPT-5, consistent with its concise style.

**Repetition and Novelty:** Measures of self-repetition and partner word reuse highlighted differences in conversational creativity. GPT-5 thinking model had much lower self-repetition and partner reuse rates, suggesting less recycling of prior phrases. This agent also scored highest in novelty, consistent with its more diverse and unpredictable language production. ChatGPT Free and Claude Opus 4.1 were moderate in both repetition and novelty.

**Questions and Interaction:** Question-asking behavior varied by agent. ChatGPT Free asked more questions on average, whereas all agents maintained a questionresponse rate of 1.0, ensuring that every question was answered. Pattern usage was negligible across agents, with only GPT-5

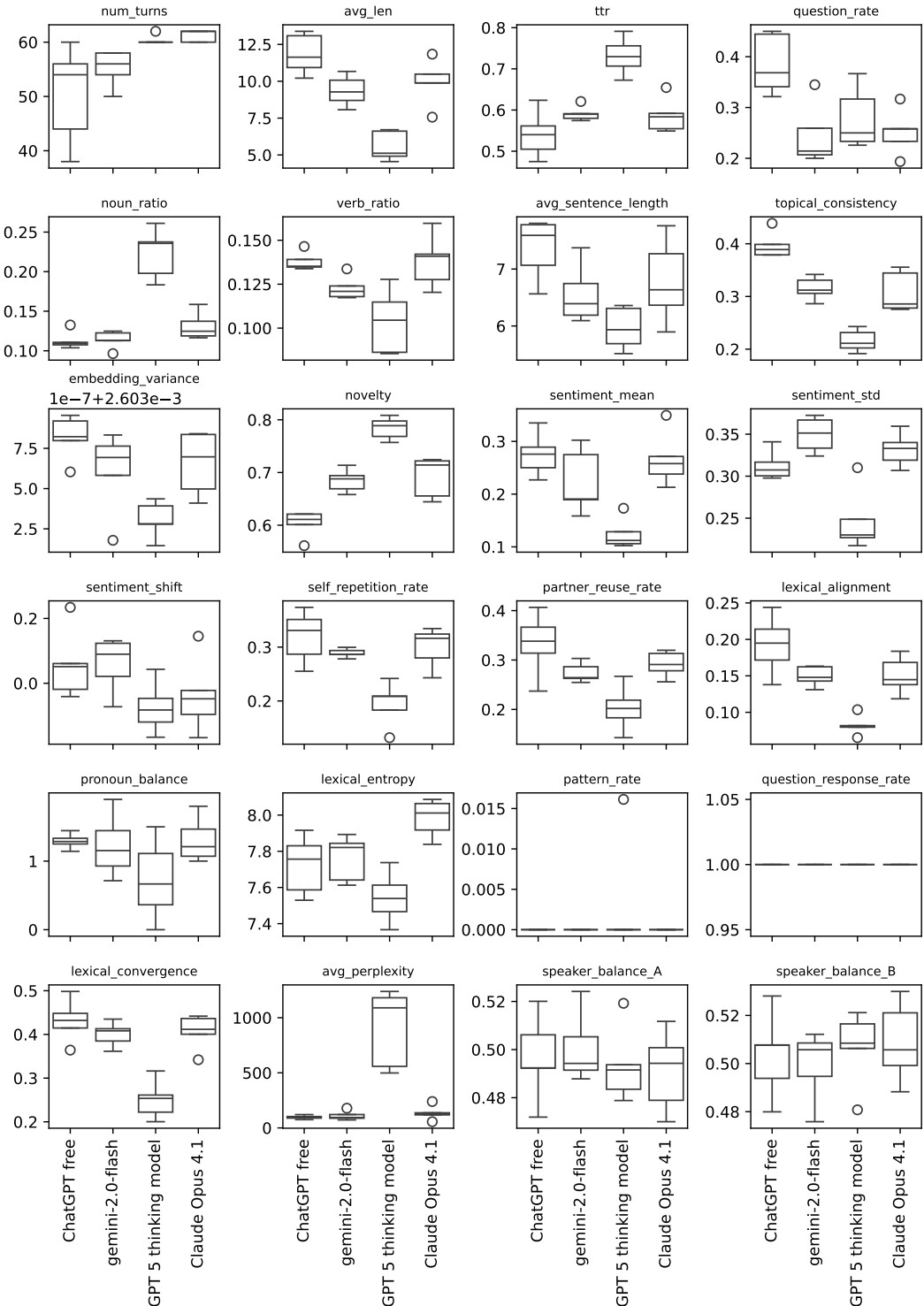

Figure 1: Box plots of each feature grouped by agent. We can qualitatively see the differences across agents.

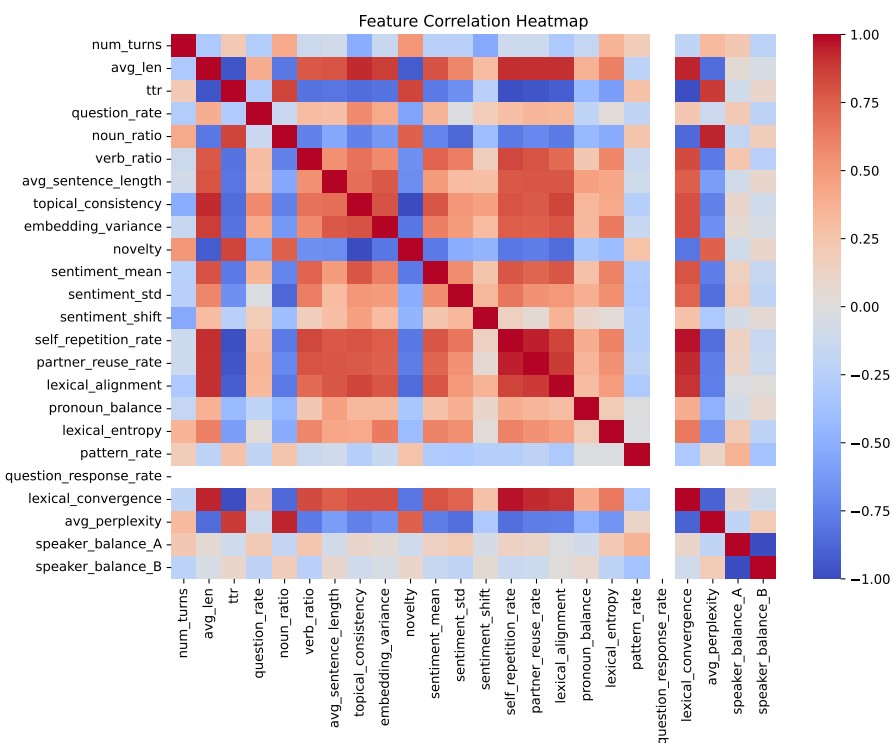

Figure 2: A heatmap of feature correlations. Several groups of highly correlated variables were identified.

showing a single outlier. Speaker balance remained around 0.5 for all agents, confirming that both participants contributed equally. GPT-5 exhibited a slightly lower pronoun balance, reflecting reduced self-referencing.

**Complexity:** Average perplexity analysis highlighted linguistic complexity. GPT-5 thinking model exhibited extreme median perplexity ($\sim$1000), indicating highly unpredictable or complex language, while other agents produced more moderate perplexity values.

Here is the summary of the characteristics described above. **GPT-5 thinking model:** High novelty, high lexical diversity, low repetition, shorter utterances, and extreme perplexity. **ChatGPT Free:** Longer utterances, higher question rate, more positive sentiment, stronger alignment with partner. **Claude Opus 4.1:** Longest conversations, balanced linguistic style, moderate novelty and entropy. **Gemini-2.0-flash:** Generally intermediate across most metrics, with less extreme behaviors.

We also visualized feature correlations in Figure 2. It revealed several groups of highly correlated variables ($|r| > 0.85$), including (i) `avg_len`, `ttr`, `topical_consistency`, `embedding_variance`, `novelty`, and several lexical reuse measures, (ii) `noun_ratio`, `sentiment_std`, and `avg_perplexity`, and (iii) `speaker_balance_A` and `speaker_balance_B`.

## 3.2 Statistical Comparison and Feature Importance Analysis

KruskalWallis tests confirmed that many features differed significantly between agents (Table 1). The most significant differences included topical consistency, novelty, average utterance length, number of turns, noun ratio, lexical alignment, sentiment standard deviation, TTR, and verb ratio. Features related to turn-taking balance, question-response adherence, and fixed conversational patterns were not significantly different. Random forest analysis identified topical consistency, novelty, noun ratio, number of turns, and lexical entropy as the most important features for distinguishing agents (Figure 3). Lexical diversity, self-repetition rate, and alignment metrics also contributed substantially, whereas pragmatic features such as speaker balance and question-response adherence had minimal

Table 1: Results of Kruskal-Wallis tests for all the features. The most significant differences included topical consistency, novelty, average utterance length, etc. Features related to turn-taking balance, question-response adherence, and fixed conversational patterns were not significantly different. Nominal p values are presented.

| feature | stat | p value |
|---|---|---|
| topical_consistency | 16.142857 | 0.001060 |
| novelty | 16.142857 | 0.001060 |
| avg_len | 14.325714 | 0.002494 |
| num_turns | 13.726408 | 0.003302 |
| noun_ratio | 13.422857 | 0.003806 |
| lexical_alignment | 13.064108 | 0.004500 |
| sentiment_std | 12.462857 | 0.005955 |
| ttr | 12.440000 | 0.006018 |
| verb_ratio | 12.165714 | 0.006837 |
| lexical_entropy | 12.028571 | 0.007286 |
| lexical_convergence | 11.891429 | 0.007764 |
| self_repetition_rate | 11.754286 | 0.008274 |
| avg_perplexity | 11.617143 | 0.008817 |
| sentiment_mean | 11.457143 | 0.009494 |
| partner_reuse_rate | 10.451429 | 0.015094 |
| embedding_variance | 10.177143 | 0.017119 |
| avg_sentence_length | 10.108571 | 0.017665 |
| question_rate | 9.100528 | 0.027984 |
| sentiment_shift | 6.508571 | 0.089325 |
| pronoun_balance | 3.439729 | 0.328664 |
| pattern_rate | 3.000000 | 0.391625 |
| speaker_balance_A | 0.782857 | 0.853563 |
| speaker_balance_B | 0.782857 | 0.853563 |
| question_response_rate | 0.000000 | 1.000000 |

importance. These results highlight that semantic, lexical, and stylistic properties are the strongest markers of agent-specific conversational behavior.

PCA using all features provided partial separation of agents (Figure 4 left). Restricting PCA to features with Kruskal-Wallis $p < 0.05$ and random forest importance $> 0.03$ enhanced clustering (Figure 4 right), suggesting that a subset of discriminative features captures the majority of agent-specific variation.

## 4 Discussions

This study systematically compared conversations generated by four large language models: Chat-GPT Free, Gemini-2.0-flash, GPT-5 thinking model, and Claude Opus 4.1. Across 23 linguistic, semantic, and interactional features, we observed clear and consistent differences in conversation style, complexity, and lexical diversity: GPT-5 thinking model emphasized novelty, lexical diversity, and unpredictability, producing shorter, highly varied utterances with low self- and partner-repetition rates. ChatGPT Free favored longer utterances, higher question frequency, and positive sentiment, with strong alignment to conversational partners. Claude Opus 4.1 generated the longest conversations, with balanced syntactic and lexical properties, moderate novelty, and high lexical entropy. Gemini-2.0-flash remained intermediate across most metrics, without extreme tendencies in any feature.

These patterns were corroborated by statistical tests, feature importance rankings, and PCA visualizations, indicating that content-related (topical consistency, novelty, lexical entropy, etc.) and interactional (lexical alignment, partner reuse rate, etc.) features reliably distinguish agent-generated conversations, while more basic features such as speaker balance and question response rate do not differ significantly among different agents.

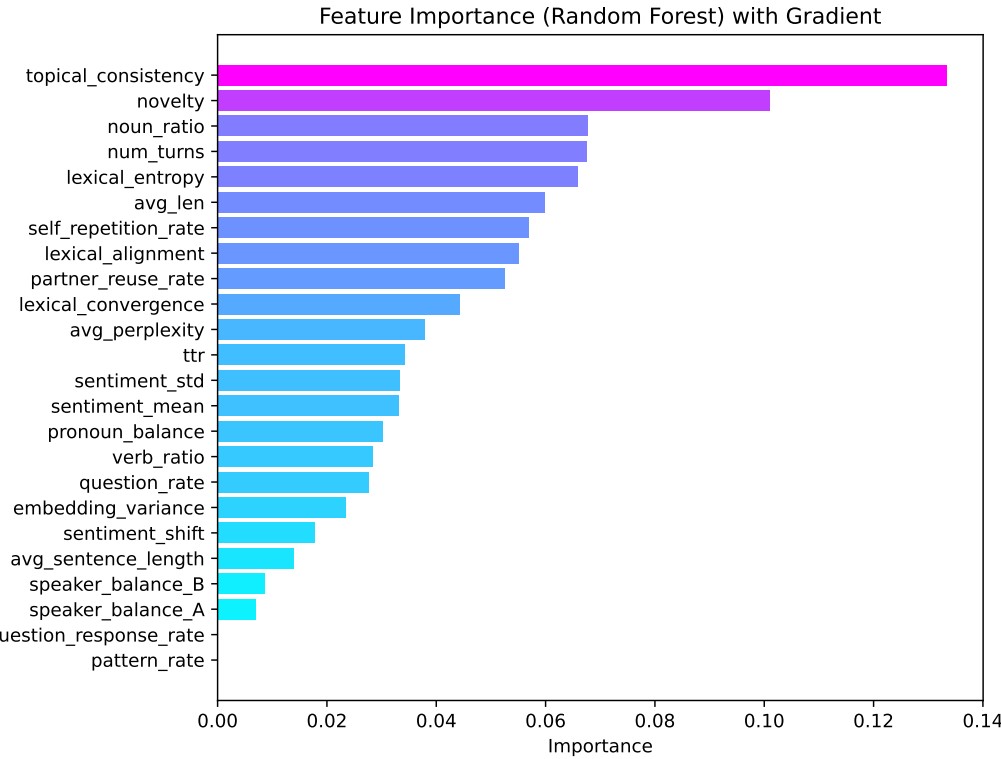

Figure 3: Important features identified by random forest analysis. Topical consistency, novelty, noun ratio, number of turns, and lexical entropy had the highest importance.

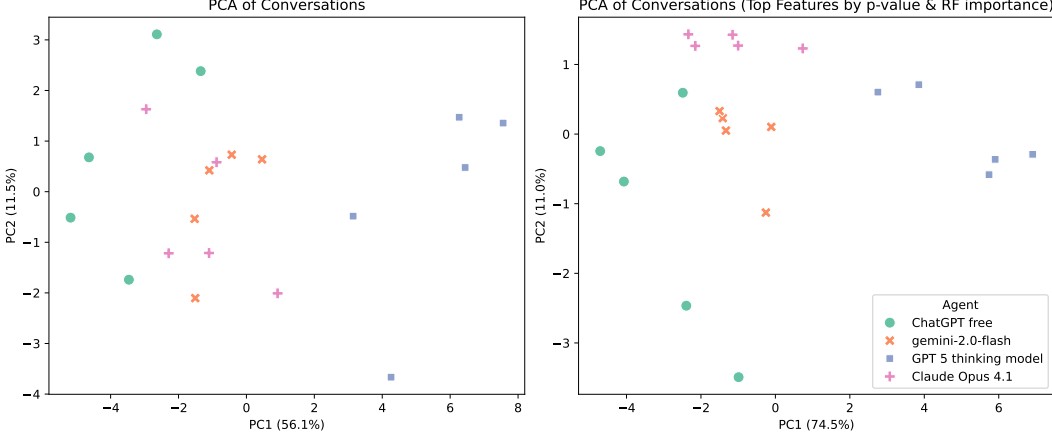

Figure 4: PCA using all features (left) and pre-selected features (right). The right PCA has an enhanced clustering compared to the left PCA.

Our results provide several key insights. Firstly, even under controlled prompts, LLMs exhibited distinct conversational fingerprints. This suggests that conversation-level metrics can serve as quantitative probes of model behavior. Secondly, features such as topical consistency, novelty, lexical entropy, and noun ratio were the most informative for distinguishing agents, whereas turn-taking and question-response adherence were uniform. This implies that traditional interaction metrics alone are insufficient to capture qualitative differences in AI-generated conversations. Thirdly, GPT-5s highly novel but unpredictable utterances highlight a trade-off between creativity and coherence. In contrast, ChatGPT and Claude maintain more predictable, partner-aligned responses, suggesting different optimization priorities in model training.

Our approach in this study systematically combines multi-turn conversation generation, quantitative feature extraction, and statistical discrimination across multiple LLMs. By providing both lexical-semantic measures and interactional metrics, this study offers a reproducible methodology for cross-agent comparison, insights into agent-specific stylistic tendencies, and a framework for evaluating novelty, complexity, and alignment in synthetic dialogues, These contributions are relevant for researchers seeking to benchmark LLMs beyond traditional metrics, and for practitioners designing conversational AI that meets specific interactional goals.

Several limitations should be noted:

- Dataset Size: Each agent generated only five conversations per prompt, which may limit generalizability. Future work could scale this to dozens or hundreds of conversations.
- Controlled Prompting: While standardized prompts enabled comparability, real-world usage involves more diverse inputs, which may alter agent behavior.
- Feature Scope: Although 23 features capture lexical, syntactic, and semantic properties, other aspects such as pragmatic reasoning, humor, or subtle discourse cues were not measured.
- Perplexity Interpretation: Extreme GPT-5 perplexity may reflect model idiosyncrasies rather than true communicative complexity; further exploration is warranted.
- Redundant Features: This was an exploratory study and there were many correlated features. These redundancy should be reduced in future studies for further analysis.

Future research could extend this framework by including human evaluation, longer dialogue horizons, and task-oriented scenarios to probe model behavior in more ecologically valid settings.

In conclusion, this study demonstrates that large language models produce quantifiably distinct conversation styles, with measurable differences in novelty, lexical diversity, sentiment, and inter-actional alignment. Our framework combining feature extraction, statistical testing, and machine learning-based discrimination provides a reproducible and interpretable approach for evaluating multi-turn AI dialogues. By highlighting the trade-offs between creativity, coherence, and alignment, these findings inform both LLM evaluation and design of more human-like conversational agents.

## Reproducibility Statement

We have placed all the data and codes to reproduce this study in the Google Colaboratory: `https://colab.research.google.com/drive/1JobFFniam7HXbCmIiBgLuLC8t5dGFTct?usp=sharing`. Readers can reproduce the whole analysis results without any environment building on their machines.

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

## Agents4Science AI Involvement Checklist

1. **Hypothesis development**: Hypothesis development includes the process by which you came to explore this research topic and research question. This can involve the background research performed by either researchers or by AI. This can also involve whether the idea was proposed by researchers or by AI.

   Answer: [C]

   Explanation: Human 1st author asked AI 1st author what research they can do without external data. AI 1st author replied multiple possibilities, among which was AI agents repeating talks. Human 1st author elaborated this into the current research question.

2. **Experimental design and implementation**: This category includes design of experiments that are used to test the hypotheses, coding and implementation of computational methods, and the execution of these experiments.

   Answer: [D]

   Explanation: Human 1st author prompted Chat GPT Free version and Gemini-2.0-flash for data generation. Human 2nd author prompted GPT-5 thinking model for data generation. Human 3rd author prompted Claude Opus 4.1 for data generation. AI 1st author suggested the list of features to evaluate. Human 1st author prompted AI 1st author to include more features, and AI 1st author suggested additional features. Visualization, Statistical analysis, and machine learning-based evaluation were all suggested by AI 1st author.

3. **Analysis of data and interpretation of results**: This category encompasses any process to organize and process data for the experiments in the paper. It also includes interpretations of the results of the study.

   Answer: [D]

   Explanation: AI 1st author wrote original codes to analyze data according to the study design. Human 1st author ran the code to generate the results. Human 1st author sent numerical results to AI 1st author and AI 1st author interpreted the results. Human 1st author sent box plot image to AI 1st author and AI 1st author described the results. Human 1st author reviewed the AI 1st author's interpretation but only fixed obvious mistakes (e.g., higher/lower).

4. **Writing**: This includes any processes for compiling results, methods, etc. into the final paper form. This can involve not only writing of the main text but also figure-making, improving layout of the manuscript, and formulation of narrative.

   Answer: [C]

   Explanation: AI 1st author drafted the Results, Discussions, and Methods section. Human 1st author converted them into LaTeX template compatible style. Human 1st author prompted AI 1st author to improve the figure layout. AI 2nd author did literature search for Introduction section prompted by Human 1st author. Human 1st author used the information to write Introduction with AI 1st author.

5. **Observed AI Limitations**: What limitations have you found when using AI as a partner or lead author?

   Description: In this study, we tried to have AI agents do as much part of research as possible. Although it allowed us to generate research paper rapidly, there were several notable limitations. Firstly, hypothesis generation by AI agents strongly depended on human prompts. Making good prompts felt harder than making relevant hypotheses without AI agents. This was also true for study design. In both cases, if prompts were not specific, AI agents tended to give very standard suggestions. Also, there were some obvious mistakes in the result interpretations, such as high vs low values. Also, AI agents sometimes generated result sentences that we did not input. For example, the PCA result interpretations were generated before we input actual results.

