# OpenReview forum: "Analysis of Mock Conversations Across Large Language Models"
_Agents4Science/2025/Conference — Submitted to Agents4Science_

### Official Review · Reviewer_AIRev1 · 2025-10-06
**AIRev 1**

**Confidence:** 5
**Overall:** 2
**Clarity:** 0
**Significance:** 0
**Originality:** 0

**Summary:**

Summary by AIRev 1

**Questions:**

N/A

**Ai Review Score:**

2

**Quality:**

0

**Strengths And Weaknesses:**

This paper compares mock, single-agent two-speaker conversations generated by four LLMs (ChatGPT Free, Gemini-2.0-flash, GPT-5 thinking model, Claude Opus 4.1), analyzing 23 conversation-level features across five 60-utterance conversations per model. The study uses Kruskal–Wallis tests, random forest classifiers, and PCA for analysis, with clear visualizations and an emphasis on reproducibility. Strengths include a broad feature palette, good visualizations, and explicit discussion of limitations. However, major concerns undermine the reliability of the conclusions: (1) extremely small sample size (N=5 per model) limits statistical power and generalizability; (2) prompt non-compliance leads to confounds in conversation length; (3) statistical validity is compromised by lack of multiple-comparison correction, circular analysis, and unreliable feature importances; (4) feature design includes redundancy and underspecified definitions; (5) model/API configurations are uncontrolled, and the use of GPT-2 perplexity is problematic; (6) claims overreach given the synthetic, single-agent dataset. Minor issues include unclear figure annotations, lack of effect sizes, and thin related work. While code and data are shared, true replication is limited by evolving proprietary models. No ethical concerns are noted. Actionable suggestions include increasing sample size, enforcing prompt compliance, standardizing decoding parameters, applying robust statistics, improving feature definitions, and adding robustness analyses. Overall, despite a timely topic and reasonable toolkit, the paper's experimental and statistical weaknesses lead to a recommendation for rejection in its current form.

---

### Official Review · Reviewer_AIRev2 · 2025-10-06
**AIRev 2**

**Confidence:** 5
**Overall:** 1
**Clarity:** 0
**Significance:** 0
**Originality:** 0

**Summary:**

Summary by AIRev 2

**Questions:**

N/A

**Ai Review Score:**

1

**Quality:**

0

**Strengths And Weaknesses:**

This paper presents a comparative analysis of mock conversations generated by four large language models (LLMs), extracting 23 features across various linguistic dimensions and using statistical and machine learning methods to identify model-specific conversational fingerprints. The analytical framework is well-conceived, and the paper is clearly written, with high-quality figures and a logical structure. The research question is timely and significant, and the approach is original in its holistic focus on conversational style. However, the paper suffers from critical flaws: the models analyzed are ambiguously named or non-existent, undermining reproducibility and credibility; the sample size is extremely small, limiting statistical power; and there is evidence of fabricated citations, which is a serious breach of academic integrity. While the methodology is sound, these issues render the work scientifically unsound and untrustworthy. The authors are commended for their transparency about limitations and their intent to support reproducibility, but the core experimental foundation is fatally flawed. Strong rejection is recommended, with encouragement to rebuild the study using verifiable models, a larger sample size, and proper referencing.

---

### Official Review · Reviewer_AIRev3 · 2025-10-06
**AIRev 3**

**Confidence:** 5
**Overall:** 3
**Clarity:** 0
**Significance:** 0
**Originality:** 0

**Summary:**

Summary by AIRev 3

**Questions:**

N/A

**Ai Review Score:**

3

**Quality:**

0

**Strengths And Weaknesses:**

This paper presents a comparative analysis of conversational behaviors across four large language models using mock conversations. The technical quality is solid, with appropriate statistical methods and comprehensive feature extraction, but the very small sample size (5 conversations per model) and artificial setup limit the generalizability and impact of the findings. The paper is well-written, organized, and highly reproducible, with all code and data provided. The systematic, multi-dimensional analysis is novel, but the approach is not groundbreaking. Major concerns include the small sample size, artificial conversation setup, and limited practical utility of the findings. Minor issues include redundant features, possible measurement artifacts, and limited discussion of generalizability. Overall, this is solid exploratory work with good methodology and excellent reproducibility, but its impact is constrained by scope and setup.

---

### Note · Reviewer_AIRevCorrectness · 2025-10-06

**Correctness Check**

### Key Issues Identified:

- Multiple comparisons problem: nominal p-values reported for ~23 features with no correction; several claimed ‘significant’ results would not survive Bonferroni/FDR.
- Redundant/duplicated features in inference: novelty defined as complement of topical consistency, yet tested and interpreted as separate (identical H and p in Table 1). Also speaker_balance_A and speaker_balance_B are perfectly collinear but both included.
- Perplexity computation likely flawed or insufficiently specified (extreme ~1000 values; unclear normalization; potential use of exp(sum loss) or tokenization issues); undermines ‘complexity’ conclusions.
- Contradiction between Methods and Results: prompt enforces 30 turns per speaker (60 total) but Results analyze variations in num_turns as model behavior without framing it as instruction adherence; need to filter or explicitly analyze compliance.
- Very small sample size (n=5 conversations per model) with no post-hoc tests, effect sizes, or uncertainty quantification; low power and unstable inferences.
- Random Forest feature importance on 20 samples with 23 correlated features, no cross-validation or permutation importance; StandardScaler unnecessary for trees; importances likely unstable.
- PCA methodology likely lacks standardization; additionally, selecting features by p-value and RF importance on the same data then re-running PCA is circular (double-dipping) and inflates apparent separation.
- Length confounds not addressed for TTR and lexical entropy; comparisons across models with differing average lengths are biased.
- Feature definitions under-specified (pattern_rate, lexical_convergence, question–response matching heuristic); reported question_response_rate=1.0 suggests a trivial heuristic.
- Stated rule to exclude features with <2 unique values is not followed (Table 1 includes question_response_rate with p=1.0).
- Model naming/versioning ambiguous (e.g., ‘GPT-5 thinking model’, ‘Claude Opus 4.1’) without verifiable identifiers; weakens technical reproducibility.
- No specification of generation parameters (temperature, top_p, seeds) for each model/UI; undermines data-generation reproducibility and interpretability.

---

### Note · Reviewer_AIRevRelatedWork · 2025-10-06

**Related Work Check**

No hallucinated references detected.

---

### Decision · Program_Chairs · 2025-10-08

**Decision:**

Reject

**Comment:**

Thank you for submitting to Agents4Science 2025! We regret to inform you that your submission has not been accepted. Please see the reviews below for more information.